# Allosteric activation of SENP1 by SUMO1 β-grasp domain involves a dock-and-coalesce mechanism

Jingjing Guo[1], Huan-Xiang Zhou[2]*

[1]Henan Engineering Research Center of Chiral Hydroxyl Pharmaceuticals, School of Chemistry and Chemical Engineering, Henan Normal University, Xinxiang, China; [2]Department of Physics, Institute of Molecular Biophysics, Florida State University, Tallahassee, United States

**Abstract** Small ubiquitin-related modifiers (SUMOs) are conjugated to proteins to regulate a variety of cellular processes. SENPs are cysteine proteases with a catalytic center located within a channel between two subdomains that catalyzes SUMO C-terminal cleavage for processing of SUMO precursors and de-SUMOylation of target proteins. The β-grasp domain of SUMOs binds to an exosite cleft, and allosterically activates SENPs via an unknown mechanism. Our molecular dynamics simulations showed that binding of the β-grasp domain induces significant conformational and dynamic changes in SENP1, including widening of the exosite cleft and quenching of nanosecond dynamics in all but a distal region. A dock-and-coalesce mechanism emerges for SENP-catalyzed SUMO cleavage: the wedging of the β-grasp domain enables the docking of the proximal portion of the C-terminus and the strengthened cross-channel motional coupling initiates inter-subdomain correlated motions to allow for the distal portion to coalesce around the catalytic center.

*For correspondence: hzhou4@fsu.edu

Competing interests: The authors declare that no competing interests exist.

## Introduction

Whereas conjugation with ubiquitin targets proteins for degradation, conjugation with small ubiquitin-related modifiers (SUMOs), or SUMOylation, is involved in various cellular processes and required for normal growth and development in all eukaryotes (*Johnson, 2004*). Although there is only 18% sequence identity between ubiquitin and SUMO1 (the founding member of the SUMO family [*Okura et al., 1996*]), their structured domains share a common fold known as β-grasp, in which five β-strands wrap around an α-helix (*Bayer et al., 1998*) (*Figure 1*). SUMOylation, like ubiquitination, is through an isopeptide linkage between a conserved Gly-Gly motif at the C-terminus of SUMOs and a lysine sidechain on target proteins. All SUMOs are translated as precursors that are first processed by the SENP family of SUMO-specific proteases, whereby a short C-terminal extension is cleaved to expose the reactive Gly-Gly motif. Like the much better-known posttranslational modification phosphorylation, SUMOylation is reversible, and de-SUMOylation is also catalyzed by SENPs (*Yeh, 2009*). The catalytic activity of SENPs on short peptide substrates is low, likely to avoid off-target cleavage, and is substantially enhanced by the β-grasp domain of SUMOs (*Mikolajczyk et al., 2007*). The mechanism of this allosteric activation has remained poorly understood. The aim of the present study was to gain insight into the allosteric mechanism through extensive molecular dynamics (MD) simulations.

The human genome encodes three SUMOs (SUMO1, SUMO2, and SUMO3) and six SUMO-specific SENPs (SENP1, SENP2, SENP3, SENP5, SENP6, and SENP7). SENPs belong to the cysteine protease superfamily, and regulate both physiological and pathological processes mediated by

**Figure 1.** Structure of the SENP1-SUMO1 complex (PDB entry 2IY1). SENP1 is shown as gray surface and cartoon representations in the left and right panels, respectively, and the closed catalytic channel is boxed and enlarged in the middle panel. Sidechains of the catalytic triad and of two tryptophans shaping the catalytic channel are shown as sticks. Three channel-lining loops (residues 464–466, 530–533, and 599–602) are shown in orange; two exosite interface regions (residues 443–453 and 496–514) are shown in mauve. SUMO1 is shown in green but with the conserved C-terminal Gly-Gly motif in red.

SUMOylated proteins (*Yeh, 2009*). As an example, hypoxia (i.e., low oxygen), a condition common to fetal development and tumorigenesis, induces the nuclear entry and SUMOylation of hypoxia-inducible factor 1α (HIF1α) (*Cheng et al., 2007*). SUMOylated HIF1α is subject to ubiquitin-dependent degradation, and hence deSUMOylation of HIF1α by SENP1 is central to hypoxia response. For this and other reasons SENP1 is overexpressed during the development of prostate cancer, and has emerged as a potential therapeutic target (*Cheng et al., 2006*; *Bawa-Khalfe et al., 2010*; *Wang et al., 2013*). Several types of SENP inhibitors have been identified, including SUMO variants or C-terminal fragments tethered with electrophilic traps (*Hemelaar et al., 2004*; *Borodovsky et al., 2005*; *Dobrota et al., 2012*) and small molecules (*Albrow et al., 2011*; *Ponder et al., 2011*; *Chen et al., 2012*; *Madu et al., 2013*). The therapeutic value of these inhibitors is limited by covalent linkage with the catalytic cysteine residue, low specificity, or low potency. Better understanding of the mechanism for the allosteric activation of SENPs may lead to new avenues for drug development.

Structural studies have shown that SENP catalytic domains only undergo localized conformational rearrangements upon binding SUMOs (as precursors, processed products, or SUMO conjugates), and that SENP-SUMO interactions are largely similar during precursor processing and deSUMOylation (*Reverter and Lima, 2004*, *2006*; *Shen et al., 2006a*, *2006b*; *Xu et al., 2006*; *Alegre and Reverter, 2014*). In SENP1, the catalytic triad comprises residues His533, Asp550, and Cys603 (*Figure 1*). The catalytic domain (residues 419–644) consists of eight α-helices and seven β-strands, divided into two subdomains. The lower subdomain contains the α1, α2, and α8 helices and the β1-β2 hairpin, while the upper subdomain contains the α4-α6 helices and the β3-β7 sheet. The two remaining helices, α3 and α7, glue the two subdomains and also form the bottom of a channel where the extended C-terminus of SUMO1 lies and the catalytic triad is situated. Three loops, including residues 464–466 between α2 and α3, residues 530–533 between β4 and β5, and residues 599–602 between β7 and α7, line the channel and likely undergo transient motions during the entrance of the SUMO C-terminus into the channel. Hereafter these loops are referred to as loopA, loopB, and loopC, respectively. One tryptophan residue, Trp534, floors the substrate while another, Trp465, forms a lid over its conserved Gly-Gly motif. The burial of the catalytic center in the channel and consequently the transient motions around the channel necessary for proper alignment of the substrate with respect to the catalytic center may contribute to the low catalytic activity of SENPs on short peptides.

The β-grasp domain of SUMO1 binds into a large cleft to the side (hereafter the exosite), and makes separate contacts with the two subdomains of SENP1: the β1-β2 hairpin in the lower subdomain and the α4-α5 helices in the upper subdomain (*Figure 1*). An insertion in the β1-β2 hairpin of

SENP2 that extended the interface with SUMO2 resulted in no other change in the structure of the catalytic domain but nevertheless increased its catalytic activity (*Alegre and Reverter, 2014*). Given that SENPs generally lack major conformational changes upon binding SUMOs, there is no simple explanation for the allosteric activation of SENPs by the β-grasp domain of SUMOs. In this context, we note that, in addition to conformational changes, the possibility that changes in conformational dynamics can mediate allosteric effects has received attention in the recent literature (*Guo and Zhou, 2016*).

Similar to SENPs, deubiquitinating enzymes (DUBs) in the cysteine protease class also have very low catalytic activity on short peptide substrates (*Dang et al., 1998*). Although the catalytic domains of DUBs in different families have different structural scaffolds, they all contain a catalytic channel that bears resemblance to that in SENPs (*Hu et al., 2002*; *Misaghi et al., 2005*; *Messick et al., 2008*). The resemblance further extends to the positioning of the ubiquitin domain in the exosite. It may thus be expected that allosteric activation mechanisms of SENPs and DUBs share certain similarities.

A major step toward elucidating the allosteric activation of SENP1 by the β-grasp domain of SUMO1 was taken in a recent study (*Chen et al., 2014*). These authors confirmed the ability of a pre-bound β-grasp domain, as contained in a truncated SUMO1 construct (residues 1–92, hereafter referred to as trunSUMO1), to enhance the SENP1 catalytic activity on a short peptide substrate, and further traced the enhancement solely to an increase in $k_{cat}$. Their NMR data showed that trunSUMO1 binding yielded a gradient in backbone amide chemical shift perturbations (CSPs) emanating from the exosite cleft to the catalytic center. Significant sidechain methyl CSPs were also observed on six residues that dotted the region from the exosite cleft to the catalytic center, which indicated to the authors that the allosteric effect was propagated through the hydrophobic core. Their Carr-Purcell-Meiboom-Gill (CPMG) relaxation dispersion data showed that trunSUMO1 binding enhanced microsecond-millisecond (μs-ms) dynamics for residues around the catalytic center, suggesting a realignment of these residues. Chen et al.'s NMR data provided residue-level information, but their sparseness precluded a full picture on how the allosteric communication occurred in SENP1 upon binding trunSUMO1.

Here we present an atomistic picture for the allosteric communication derived from MD simulations. The simulations showed that trunSUMO1 binding induces significant conformational and dynamic changes in SENP1, including widening of the exosite cleft and quenching of nanosecond (ns) dynamics in all but a distal region. Calculated backbone amide and sidechain methyl CSPs are in broad agreement with the experimental data of *Chen et al. (2014)* but more pronounced, and our sidechain CSPs clearly identify two hydrophobic pathways, each within a subdomain, for allosteric communication from the exosite cleft to the catalytic center. The β-grasp domain, by serving as a bridge that links the two exosite interface regions, strengthens intra- and inter-subdomain motional coupling, which in turn may be the underlying reason for the quenching of ns dynamics and potentially may also be the instigator of inter-subdomain μs-ms dynamics (*Guo et al., 2015*). The concerted action of the β-grasp induced conformational and dynamic changes is captured by a dock-and-coalesce mechanism for SENP-catalyzed SUMO cleavage, whereby the wedging of the β-grasp domain into the exosite cleft enables the docking of the proximal portion of the C-terminus, and the strengthened cross-channel coupling initiates inter-subdomain correlated motions to allow for the distal portion to coalesce around the catalytic center.

## Results

We carried out three replicate explicit-solvent MD simulations for apo SENP1, for SENP1 bound with trunSUMO1, and for SENP1 bound with the full-length SUMO1 precursor (referred to as pre-SUMO1 hereafter). For each system, the three simulations accumulated 1.7 μs of total time and are denoted as sim1, sim2, and sim3.

### β-Grasp binding induces wider exosite cleft and stronger inter-subdomain contact

To compare the conformational sampling by the apo, trunSUMO1-bound, and preSUMO1-bound forms of SENP1, we carried out principal component analysis on the replicate simulations of the three systems. The distributions of conformations in the plane of the first two principal components

(PC1 and PC2), translated to free energy surfaces according to the Boltzmann relation, are displayed in *Figure 2—figure supplement 1a*. The three systems cover overlapping as well as distinct regions in conformational space. The apo form samples disconnected free energy basins, in contrast to the two bound forms, indicating a decrease in flexibility upon β-grasp binding (see below).

Both PC1 and PC2 feature prominent displacements in the two exosite interface regions and in the loops lining the catalytic channel (*Figure 2—figure supplement 1b,c*). Relative to the apo form, the two bound forms move along positive PC1 and PC2, both of which involve the moving apart between the two interface regions and the opposite movements of loopA and loopB with respect to the β-grasp domain. We hence directly monitored the relative motions between the two interface regions and between the three channel-lining loops (*Figure 2*). The distributions of the cleft distance, defined as between the centers of heavy atoms of the interface residues 448–453 and 506–513 (shown as mauve in *Figure 2a*), are displayed in *Figure 2b* for the three forms of SENP1. The mean and standard deviation of the cleft distance change from 22.4 ± 1.5 Å for the apo form to 23.1 ± 0.5 Å and 22.6 ± 1.5 Å for the trunSUMO1-bound and preSUMO1-bound forms, respectively, indicating widening of the exosite cleft upon β-grasp binding (*Figure 2—figure supplement 2a,b*).

To characterize the movements of the three channel-lining loops, we defined a coordinate system attached to the α3 and α7 helices, which form the bottom of the channel (*Figure 2a*). In the coordinate system, the z axis is along the helical axis of α7 (as defined by the vector from the Cα center of residues 612–615 to the Cα center of residues 604–607); the x axis goes through the Cα center of α3 residues 469–480 and hence points to the exosite cleft; and the y axis points into the upper subdomain. None of the three loops exhibits any overt movement along z, so we focus on the differences among the three forms of SENP1 in the distributions of the Cα center x and y coordinates of the three loops (*Figure 2c* and *Figure 2—figure supplement 2a,b*). Relative to the apo form, the two bound form show similar movements for both loopA and loopB, the former away from the exosite cleft (i.e., decreasing x) and toward the upper subdomain (i.e., increasing y) while the latter toward the cleft and away from the channel (i.e., increasing x and y); loopC moves in opposite directions in the two bound forms but perhaps that is accidental due to the long distance from the binding interface. The movement of loopA (toward α7) leads to stronger inter-subdomain contact; the x component of loopB's movement has a similar effect, while the y component may create space for the docking of the proximal portion of a C-terminus when tethered to the β-grasp domain.

The stronger inter-subdomain contact induced by β-grasp binding can be further illustrated by the movement of the sidechain of the channel lid residue, W465 (*Figure 2d*). In apo SENP1 this sidechain samples a broad range of positions, but in preSUMO1-bound SENP1 it stays stably near the top of α7 (on the sub-μs timescale); the situation is intermediate in trunSUMO1-bound SENP1.

## Nanosecond dynamics of SENP1 is quenched upon β-grasp binding

As noted above, the narrowing of accessible conformational regions upon β-grasp binding suggests a decrease in flexibility, i.e., quenching of ns dynamics. To compare the residue-specific flexibilities among the three forms of SENP1, we calculated their Cα root-mean-square fluctuations (RMSFs). Relative to apo SENP1, the two bound forms both show decreased flexibility throughout most of the amino acid sequence, except for the N- and C-terminal segments (*Figure 3a*).

To visualize how the changes in RMSF are distributed spatially, we display them according to a color scale on the structures of the two bound forms (*Figure 3b,c*). In both systems, rigidification propagates from the exosite cleft to the entire upper subdomain and to most of the lower subdomain. The rigidification is compensated to some extent by higher flexibility in the distal region of the lower subdomain, comprising the N- and C-terminal segments. Therefore, the allosteric effects elicited by the β-grasp domain include both the stronger inter-subdomain contact and the quenching of ns dynamics in all but a distal region.

## Calculated sidechain methyl CSPs identify two hydrophobic pathways for allosteric communication

*Chen et al. (2014)* measured the backbone amide CSPs of SENP1 upon trunSUMO1 binding. Their CSPs can be described as a gradient emanating from the exosite cleft to the catalytic center (*Figure 4a*), a pattern that is somewhat similar to that of the changes in RMSF (*Figure 3b*). In addition, they observed significant sidechain methyl CSPs on six residues, including Leu450, Val501, Val

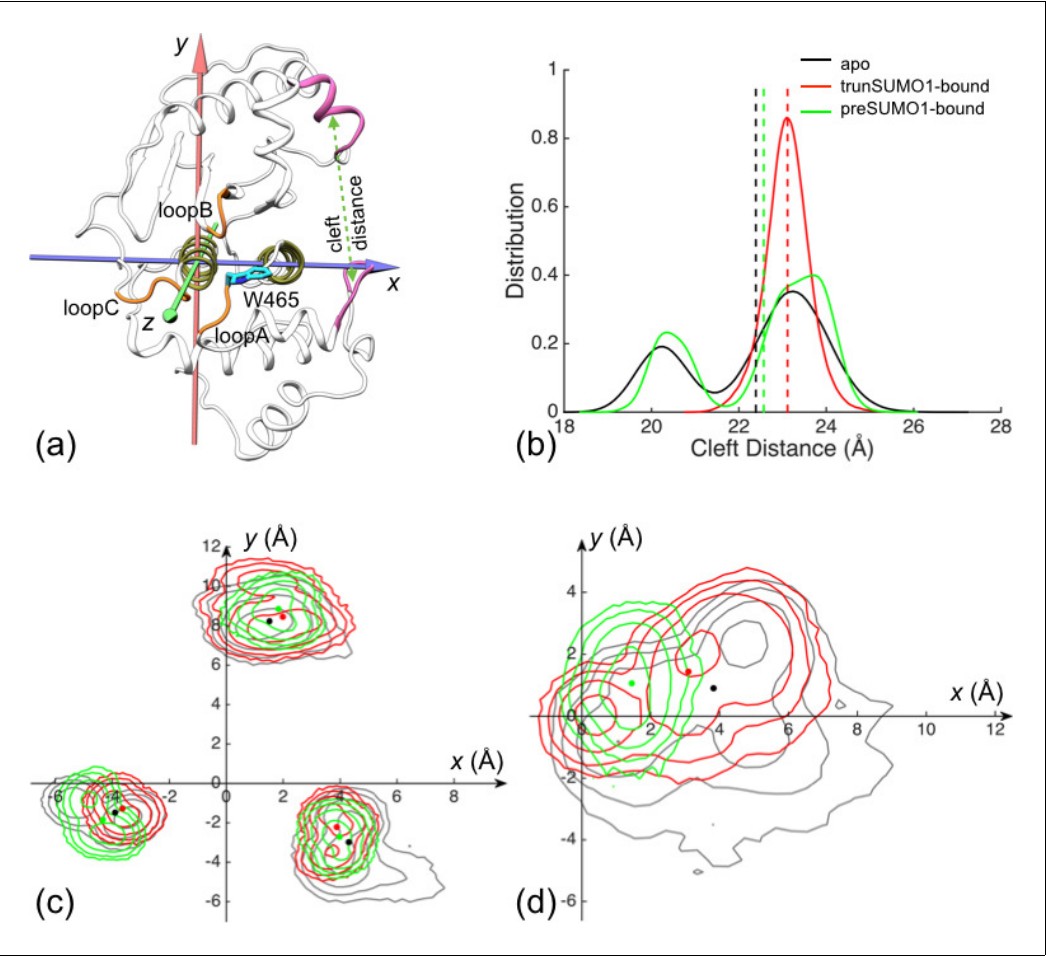

**Figure 2.** The displacements of the exosite interface regions and channel-lining loops upon SUMO1 binding. (**a**) The regions (displayed in mauve) used for defining cleft distance (indicated by double-headed arrow) and the coordinate system used for defining movements of the three channel-lining loops (displayed in orange) and the W465 sidechain (displayed with carbon atoms in cyan). (**b**) The distributions of the cleft distance in the simulations of the apo and trunSUMO1- and preSUMO1-bound forms of SENP1. The average value of each system is shown as dash with matching color. (**c**) Distributions of the Cα centers of the three loops in the x-y plane. The average positions of the loops in each system are shown as dots. (**d**) Corresponding results for the center of W465 sidechain heavy atoms.

The following figure supplements are available for figure 2:

**Figure supplement 1.** Difference in conformational sampling among apo SENP1 and two SUMO1-bound forms.

**Figure supplement 2.** Comparison of representative structures from replicate simulations of different systems.

---

509, V516, Val518, and ValV532, which sparsely occupy the region from the exosite cleft to the catalytic center (*Figure 4a*).

To compare with these NMR results, we calculated the chemical shifts of backbone amides and sidechain methyls on the apo and trunSUMO1-bound SENP1 simulations. The resulting CSPs are displayed in *Figure 4b*. Overall, they agree well with the experimental data, but are more pronounced. The calculated backbone amide CSPs spread all the way to the back, in both subdomains. The calculation identifies three of the six experimentally detected residues as well as 12 others as having significant sidechain methyl CSPs (>0.08 ppm). Interestingly, with the 12 additional residues, two hydrophobic pathways emerge, each connecting an exosite interface region in one subdomain to the catalytic center. Chemical shifts are a measure of both conformational and dynamics effects.

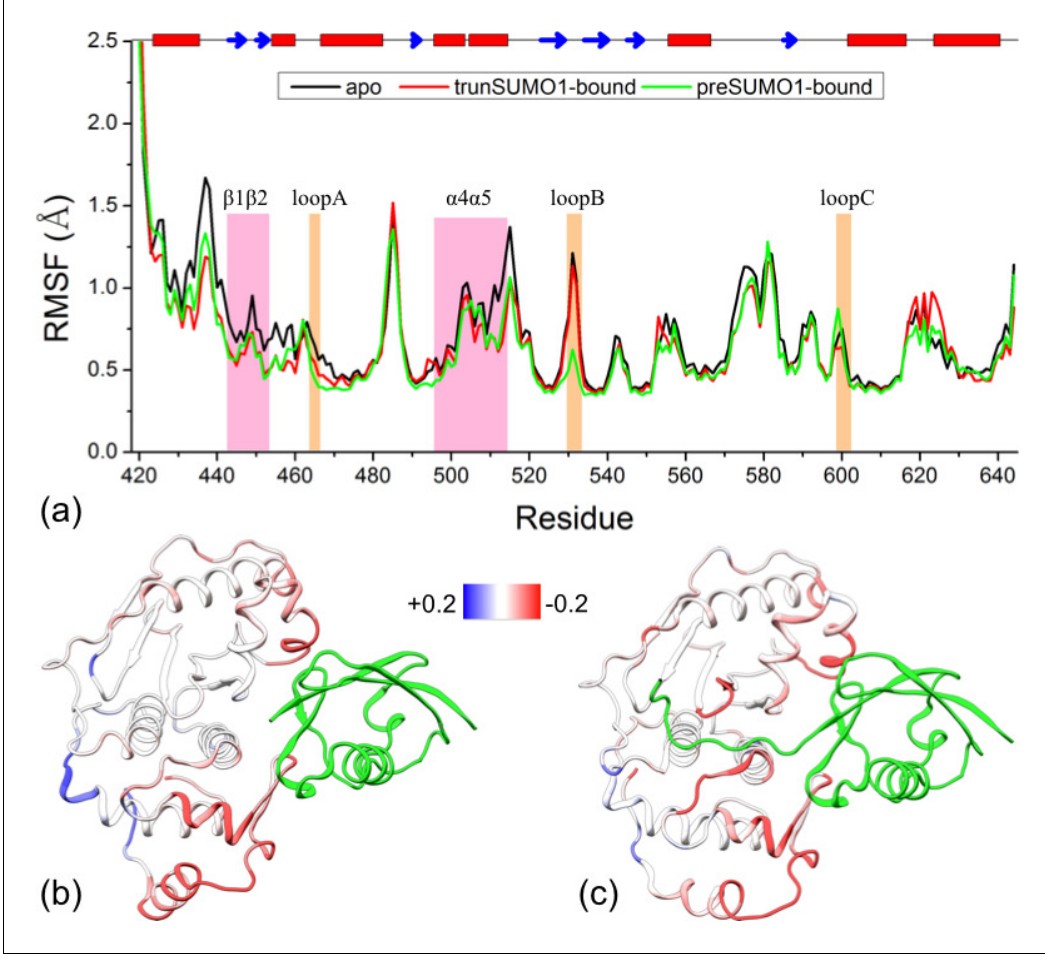

**Figure 3.** Comparison of flexibility among the three systems, as measured by Cα atom root-mean-square fluctuations (RMSFs). (a) Variations of RMSF along the amino acid sequence for the apo, trunSUMO1- and preSUMO1-bound forms of SENP1. The two exosite interface regions and three channel-lining loops are highlighted by shading in mauve and orange, respectively. (b–c) Changes in RMSF upon binding trunSUMO1 and preSUMO1, displayed on the bound structures according to a color scale (shown; red and blue corresponding to lower and higher flexibilities, respectively).

Hence the sidechain methyl CSPs are a manifestation of the conformational and dynamic changes induced by β-grasp binding, and the hydrophobic pathways may represent short paths over which allosteric effects are propagated from the exosite cleft to the catalytic center.

## Strengthened intra- and inter-subdomain motional coupling underlies quenching of ns dynamics

Community analysis is a way to reveal the pattern of motional coupling within a protein, based on residue-residue physical contact and positional correlation during an MD simulation (*Sethi et al., 2009*). A protein structure is partitioned into communities, within which residues form dense contacts but between which residues form sparse contacts. The strength of coupling, or betweenness, between two communities is determined by the magnitudes of positional correlations within networks of contacting residues.

Results of our community analysis are displayed in *Figure 5* for apo and trunSUMO1-bound sim1, and in *Figure 5—figure supplement 1* and *Figure 5—figure supplement 2* for the other two replicate simulations. Very similar community structures are obtained from the replicate simulations of each system, so here we focus on sim1. In all cases, five major communities (numbered 1 to 5) can be recognized for SENP1, anchored by **β1β2** and **α2; α3; α4α5; α6** and various parts of β4–β7; and

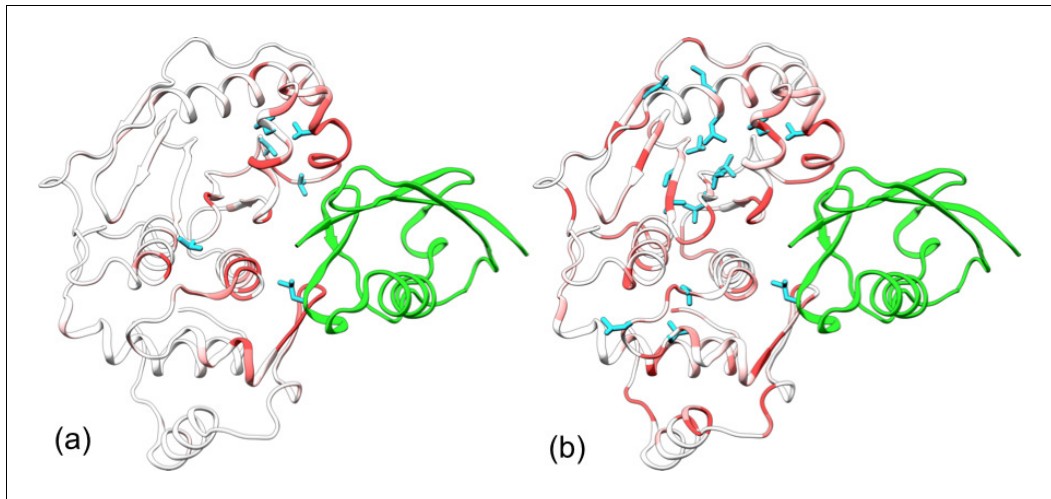

**Figure 4.** Comparison of experimental and calculated chemical shift perturbations (CSPs) of SENP1 upon trunSUMO1 binding. (a) NMR results of *Chen et al. (2014)*. (b) Calculated results. Backbone amide CSPs are displayed according to a color scale (red to gray corresponding to high to low CSPs); sidechain methyls with significant CSPs (>0.05 in panel a and >0.08 in panel b) are shown as cyan sticks.

**α7**, respectively, and possibly containing other segments. In the apo form, in addition to these five major communities, four minor communities (numbered 1′, 2′, 4′, and 4″) are also formed. In the trunSUMO1-SENP1 complex, SENP1 is partitioned only to the five major communities, with the minor communities absorbed. In particular, communities 4, 4′, and 4″ in apo SENP1 coalesce into a single community upon trunSUMO1 binding.

Moreover, the inter-community betweennesses are also strengthened, especially between the two communities, 2 and 5, anchored by the two central helices (**α3** and **α7**, respectively) and community 1 in the lower subdomain and community 4 in the upper subdomain. This strengthened inter-community coupling is in line with the stronger inter-subdomain contact noted above. It comes about because the β-grasp domain couples strongly to both subdomains. Therefore, the β-grasp domain, by serving as a bridge linking the two exosite interface regions, strengthens the motional coupling both within and between the subdomains of SENP1. A similar bridging role was identified for a peptide in inducing inter-domain allosteric communication in Pin1 (*Guo et al., 2015*).

With the strengthened intra-subdomain coupling upon trunSUMO1 binding, it can be expected that allosteric communication from the exosite cleft to the catalytic center becomes more effective. The effectiveness of allosteric communication between two sites can be measured by the lengths, defined using the residue-residue positional correlations (i.e., stronger correlations correspond to shorter lengths), of paths connecting the sites. Indeed, the path lengths from the two exosite interface regions to the catalytic center are shorter in the trunSUMO1-bound form than in the apo form. This result is in line with the two hydrophobic pathways identified by sidechain methyl CSPs.

The simultaneous occurrence of strengthened intra- and inter-subdomain coupling and quenching of ns dynamics observed here on SENP1 upon binding an allosteric activator conforms to a pattern previously recognized from a number of other proteins (including Pin1) in which conformational dynamics plays a prominent role in mediating allosteric communication (*Guo and Zhou, 2015*). This pattern was explained by a dynamic model of allostery. According to this model, fast motions (e.g., those on a ns timescale) are uncorrelated; an allosteric activator strengths inter-community coupling, which in turn leads to quenching of fast motions. The strengthened coupling here comes about because the β-grasp domain serves as a bridge that links the two interfacial regions. The model further predicts another dynamic effect, i.e., slower (e.g., μs-ms), cross-community correlated motions are initiated. Indeed, Chen et al.'s CPMG data for residues around the catalytic center indicated enhanced μs-ms conformational exchange upon trunSUMO1 binding (*Chen et al., 2014*).

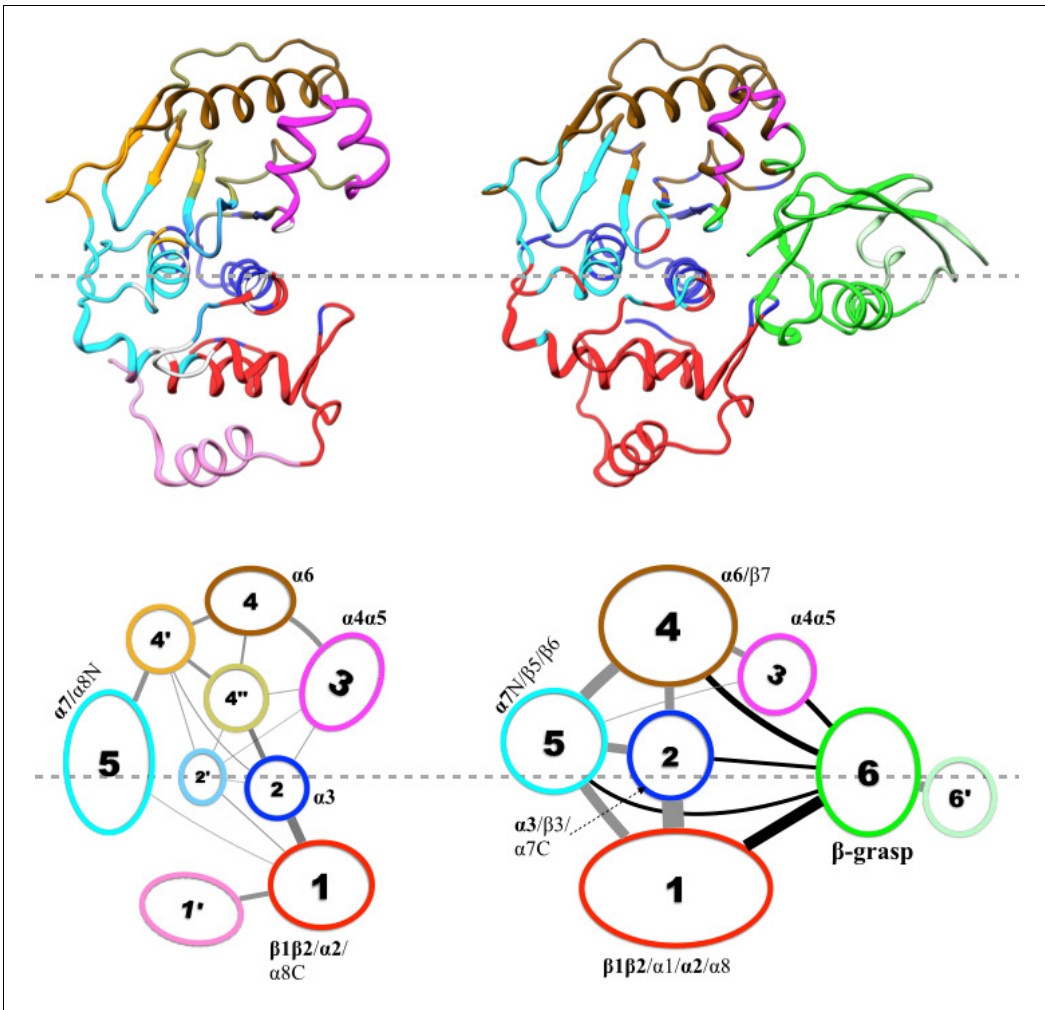

**Figure 5.** Results of community analysis for apo and trunSUMO1-bound SENP1 sim1, displayed on the left and right panels, respectively. Communities are displayed either by different colors on the structures (upper row) or as numbered ovals with matching colors (lower row). In the lower row, inter-community cumulative betweennesses are displayed by the thickness of the lines connecting communities. The community analysis was performed using the NetworkView plugin in VMD (*Sethi et al., 2009*).

The following figure supplements are available for figure 5:

**Figure supplement 1.** Results of community analysis for apo and trunSUMO1-bound SENP1 sim2.

**Figure supplement 2.** Results of community analysis for apo and trunSUMO1-bound SENP1 sim3.

## Allosteric effects can be reproduced by restraining the two exosite interface regions at an widened separation

The results presented above lead to the hypothesis that the conformational and dynamic effects of β-grasp binding are achieved through increasing the exosite cleft distance and then rigidly holding the two interface regions at the widened separation. As demonstrated previously for Pin1 (*Guo et al., 2015*), this type of hypotheses can be tested by restrained MD simulations. Specifically, we ran an apo simulation but restrained the two exosite interface regions (i.e., the β1β2 hairpin in the lower subdomain and the α4α5 helices in the upper subdomain) to their conformation from a snapshot in trunSUMO1-bound sim1, in which the exosite cleft distance is at a widened 23.9 Å (*Figure 6a*).

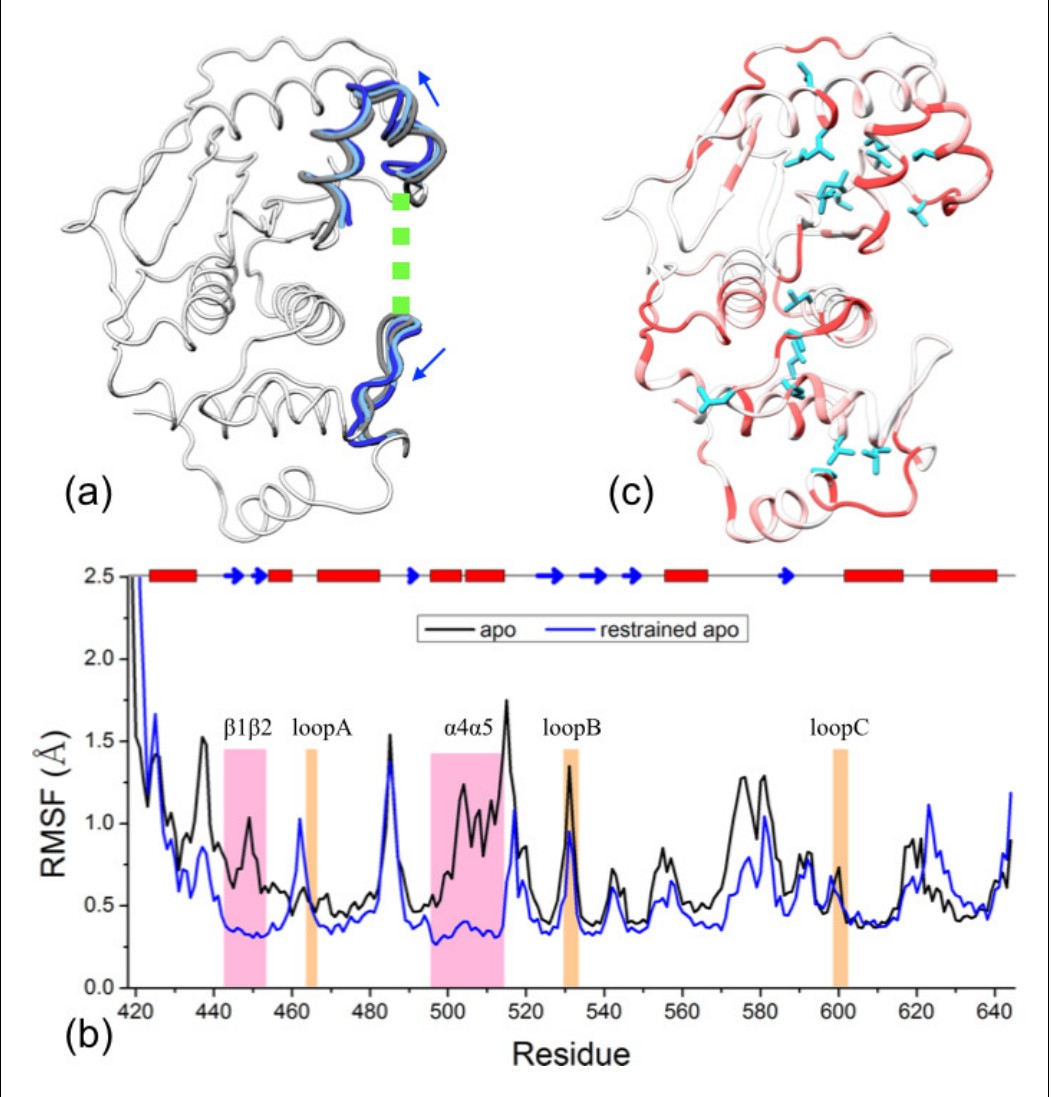

**Figure 6.** Reproduction of allosteric effects by restraining the exosite interface regions of apo SENP1 at a widened separation. (a) Illustration of the restraint. Blue arrows indicate the widening of the exosite cleft, and green dash indicates the subsequent restraint. (b) Comparison of RMSFs between apo sim1 and the restrained apo simulation. (c) Backbone amide and sidechain methyl CSPs of the restrained simulation (to be compared with *Figure 4b*).

The following figure supplement is available for figure 6:

**Figure supplement 1.** The displacements of the channel-lining loops for a control simulation in which the exosite interface regions of apo SENP1 are restrained at a widened separation.

---

The simulation of this control system reproduces the main conformational and dynamic effects of β-grasp binding. The three channel-lining loops in the control system move in essentially the same ways as those in the trunSUMO1-bound form (compare *Figure 6—figure supplement 1a,b* with *Figure 2c* and *Figure 2—figure supplement 2b*). The RMSF of the control system (*Figure 6b*) also shows a similar albeit slightly more pronounced decrease, indicating quenching of ns dynamics, for most of the SENP1 structure, including all the three channel-lining loops. Lastly the calculated backbone amide CSPs of the control system have a similar gradient to that of the trunSUMO1-bound form, and the calculated sidechain methyl CSPs again identify two hydrophobic pathways that may propagate allosteric effects from the exosite cleft to the catalytic center (*Figure 6c*).

## Discussion

Through extensive MD simulations, we have found that SUMO1 β-grasp binding to SENP1 induces significant conformational and dynamic effects, including the widening of the exosite cleft and the quenching of ns dynamics in all but a distal region. The calculated CSPs are in broad agreement with the NMR data of *Chen et al. (2014)* but our results overall present a deeper, atomistic picture for the allosteric communication from the exosite cleft to the catalytic center. We conclude that the wedging into the exosite cleft and the bridging between the two exosite interface regions by the β-grasp domain are the underlying reason for the widened exosite cleft and the quenched fast dynamics, and may also be the instigator of inter-subdomain slow motions. Enhanced μs-ms conformational exchange has indeed been observed in CPMG experiments on trunSUMO1-bound SENP1 (*Chen et al., 2014*). Our findings have broad implications for the mechanism of allosteric activation of SENPs by SUMOs, the SUMO paralogue specificity of SENPs, and the development of drugs targeting SENPs.

### A dock-and-coalescence mechanism for SENP-catalyzed SUMO cleavage

The concerted action of the β-grasp induced conformational and dynamic changes is expected to help both the binding step and the subsequent catalytic step for trunSUMO1-bound SENP1 reacting with an isolated peptide substrate. First, widening of the exosite cleft may enhance the rate at which the substrate binds to the catalytic center, although this effect may not be detectable in an enzymatic assay if the binding does not rate-limit the overall enzymatic reaction. Second, initiation of slow conformational dynamics may facilitate the proper alignment of the substrate around the catalytic residues. These putative effects can explain the observation of *Chen et al. (2014)* that trunSUMO1 enhances the SENP1 catalytic activity on a peptide substrate through increasing $k_{cat}$.

In SENP-catalyzed SUMO cleavage for processing of SUMO precursors and de-SUMOylation of target proteins, the C-terminal substrate is tethered to the β-grasp domain via an extended linker. Tethering can also help both the binding step and the subsequent catalytic step. The binding of a substrate-containing SUMO (i.e., a precursor or SUMO conjugate) likely occurs in a sequential manner: the β-grasp domain first wedges into the exosite cleft and the C-terminus then docks into the catalytic channel. Tethering can facilitate this binding by correctly orienting the C-terminal substrate. This effect and the β-grasp induced widening of the exosite cleft together may speed up the docking of the C-terminus into the catalytic channel sufficiently as to make the full binding step rate-limited by the initial wedging of the β-grasp into the exosite. Tethering can also reinforce the motional coupling between the exosite interface regions and the channel-lining loops. This may in turn intensify the slow conformational dynamics that facilitates the proper alignment of the substrate around the catalytic residues, thereby accelerating the catalytic step.

In previous studies we have put forward a docking-and-coalesce mechanism to describe the binding of intrinsically disordered proteins, which, similar to the disordered SUMO C-terminus, usually form extended conformations on their targets (*Qin et al., 2011*; *Zhou et al., 2012*). Here we adapt this mechanism to describe the steps of SENP-catalyzed SUMO cleavage reaction just before the actual cut of the (iso)peptide bond (*Figure 7*). The process starts with the wedging of the SUMO β-grasp domain into the SENP exosite cleft. This wedging widens the separation between and also tightly links the two exosite interface regions. The widening of the exosite cleft enables the docking of the proximal portion of the C-terminus into the catalytic channel. Moreover, the wedged β-grasp domain strengthens the motional coupling within and between the two subdomains of SENP, and the newly docked proximal portion of the SUMO C-terminus reinforces this coupling. According to a theoretical model (*Guo and Zhou, 2015*), the strengthened coupling sets up the condition for the emergence of inter-subdomain correlated motions, which finally allow for the distal portion of the SUMO C-terminus to align properly around the catalytic center for bond cleavage.

The preceding dock-and-coalesce mechanism differs from the previous generic version (*Qin et al., 2011*; *Zhou et al., 2012*) by the cooperation between a structure domain (i.e., the β-grasp domain) and a disordered region (i.e., the C-terminus) of the substrate protein and by the prominent role of allosteric communication within the target protein. In addition, the end result here is the transition-state complex for a catalytic reaction as opposed to a ground-state complex. Certain aspects of the mechanism presented here have been suggested previously by *Shen et al. (2006a)*, including the stimulation of catalytic channel opening by SUMO binding and the facilitation

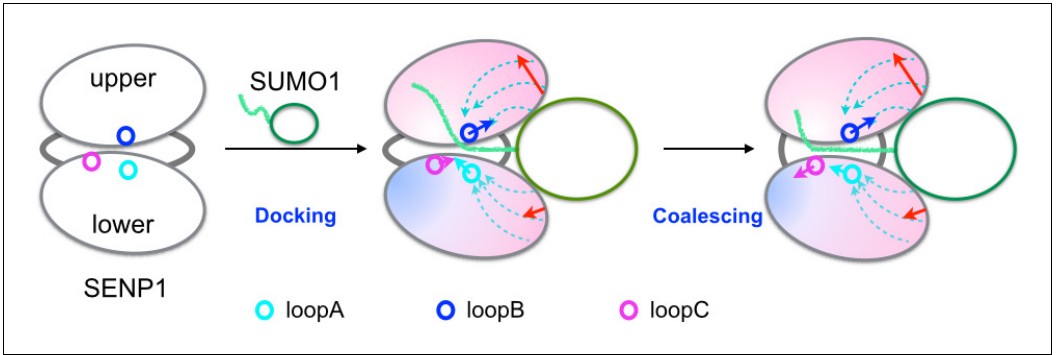

**Figure 7.** Illustration of the dock-and-coalesce mechanism for SENP-catalyzed SUMO C-terminal cleavage. SENP is shown as two ovals (representing two subdomains) connected by two strings, with the three channel-lining loops highlighted as small circles; SUMO is shown as an oval (representing the β-grasp domain) with a tail (the C-terminus). In the docking step, as the β-grasp domain wedges into the exosite cleft, the cleft separation widens (indicated by red arrows), loopA and loopB move (cyan and blue arrows) to make stronger inter-subdomain contact and also to create space for docking the proximal portion of the SUMO C-terminus; the two SENP subdomains lose flexibility on the fast timescale except for a distal region (indicated by red to blue shading), and finally the proximal portion of the SUMO C-terminus docks into the catalytic channel. Two hydrophobic pathways (bundles of dashed arrows) propagate the allosteric effects from the exosite interface regions to the catalytic center. In the coalescence step, the wedged β-grasp domain and the docked C-terminus cooperate to reinforce allosteric effects, initiating inter-subdomain correlated slow motions to allow for proper alignment of the substrate around the catalytic center.

of substrate alignment by cross-channel dynamics, without elaboration. Lastly the dock-and-coalesce mechanism may not only provide a qualitative description of the individual steps but also serve as a framework for quantitative calculations of kinetic rate constants (see next).

## Contributions of exosite SENP-SUMO interactions to paralogue specificity

The catalytic activities of a particular SENP can vary toward different SUMO paralogues (*Reverter and Lima, 2006*; *Shen et al., 2006a*; *Hickey et al., 2012*). For example, SENP1 processes preSUMO1 and deconjugates SUMO1 and SUMO2 conjugates with similar rates, but processes the SUMO2 precursor with a 25-fold lower $k_{cat}$ (*Shen et al., 2006a*). The decrease in $k_{cat}$ was attributed to a compromised fit between the C-terminal extension and the catalytic channel. On the other hand, in either processing or deconjugation, SENP2 discriminates between SUMO1 and SUMO2 not by $k_{cat}$ but by $K_M$ (*Reverter and Lima, 2006*). An approximately 10-fold lower $K_M$ for SUMO2 was attributed to an extended exosite interface.

The dock-and-coalesce mechanism presented above now allows us to more clearly delineate the contributions of exosite interactions to paralogue specificity. Two consequences can be expected of the more extensive exosite interactions of SENP2 with SUMO2 than with SUMO1. First, stronger electrostatic attraction across the exosite interface quickens the initial wedging of the SUMO2 β-grasp domain into the exosite and slows down the reverse process. Second, the more extensive exosite interactions likely cause stronger allosteric effects of the wedged β-grasp domain, resulting in faster docking of the SUMO2 C-terminus into the catalytic channel. These two consequences together may explain the lower $K_M$ for SUMO2 than for SUMO1. A precedent of the second consequence occurred in Pin1, where peptides bound at an inter-domain exosite were identified to serve a bridging role similar to the one proposed here for the β-grasp domain; a peptide with more extensive across-domain interactions indeed induced stronger allosteric effects (*Guo et al., 2015*; *Guo and Zhou, 2015*).

Here the expectation for stronger allosteric effects is specifically supported by the observation that a pre-bound SUMO2, but not SUMO1, β-grasp domain enhanced the SENP2 catalytic activity on a peptide substrate (*Mikolajczyk et al., 2007*). A similar causal link between strengthened exosite interactions and strengthened allosteric effects may be at play for a SENP2 mutant that was

designed by grafting an insertion in the β1-β2 hairpin from SENP6 (*Alegre and Reverter, 2014*). The insertion extended the exosite lower interface and increased the proteolytic activity of SENP2 for some SUMO2 conjugates.

### SENP exosite as target site for drug development

Inhibitors that target the catalytic center of SENPs have limited therapeutic value due to covalent linkage with the catalytic cysteine residue, low specificity, or low potency. The foregoing discussion suggests that the exosite may be another potential target site for drug development. Suppression and elevation of SENP activities may both be desired (under different pathological conditions) and can be achieved by disrupting and strengthening exosite SENP-SUMO interactions, respectively. Small molecules that bind at sites deeply into the SENP-SUMO interface can have a disruptive effect (*Kumar and Zhang, 2013*), whereas those bind over both SENP and SUMO can lock and strengthen their interactions.

## Materials and methods

### Molecular dynamics simulation protocols

MD simulations were carried out for three systems: apo SENP1, SENP1-trunSUMO1 complex, and SENP1-preSUMO1 complex. The starting structures of the first and third systems were from PDB entries 2CKG (*Shen et al., 2006b*) and 2IY1 (*Shen et al., 2006a*), respectively; the latter upon removing the C-terminal 9 residues starting from Glu93 was used as the starting structure of the SENP1-trunSUMO1 complex.

Three replicate simulations were performed for each system. sim1 and sim2 were performed in NAMD 2.9 (*Phillips et al., 2005*) using the CHARMM 36 force field (*Brooks et al., 2009*; *Vanommeslaeghe et al., 2010*), while sim3 was carried out in AMBER with the AMBER99SB force field (*Hornak et al., 2006*). Each system was placed in a water box with a 10-Å buffer zone. Appropriate numbers of $Na^+$ and $Cl^-$ were added to neutralize proteins and produce a NaCl concentration of 50 mM. For the NAMD simulations, the van der Waals cut-off distance was 12 Å with a switching distance of 10 Å; for AMBER simulations, the cutoff distance for non-bonded interactions was 10 Å. The particle mesh Ewald was used for computing the Coulomb interactions under the periodic boundary condition.

Before starting a simulation, the solvated system was energy minimized at three stages, first with either the whole protein molecule(s) or backbone atoms restrained and then without any restraint. After a 50-ps equilibration, the simulation was continued at constant pressure (1 bar) and constant temperature (300 K), at a 2-fs timestep.

The simulation times of sim1, sim2, and sim3 were 300, 400, and 1000 ns, respectively, for each system. sim2 and sim3 were performed using GPU acceleration. The last 150 ns (saved at 2-ps intervals) of sim1 and sim2 and last 300 ns (saved at 5-ps intervals) of sim3 were used for analyses.

One more simulation, of a control system, was also carried out. This started from the snapshot at 200 ns of apo sim1, but with the heavy atoms of the interface regions, β1β2 (residues 443–453) and α4α5 (residues 496–514), restrained to their conformation in the snapshot at 200 ns of trunSUMO1-bound sim1. The force constant was 2 kcal/Å$^2$ for each restrained atom.

### Calculation of chemical shift perturbations

The SHIFTX2 program (*Han et al., 2011*) was used to calculate diamagnetic $^1H$, $^{13}C$ and $^{15}N$ chemical shifts on SENP1 coordinates sampled from the replicate simulations for each system (see next). The CSPs of backbone amides and sidechain methyls were calculated as $\sqrt{(0.154 \cdot \Delta\delta_N)^2 + (\Delta\delta_H)^2}$ and $\sqrt{(0.341 \cdot \Delta\delta_C)^2 + (\Delta\delta_H)^2}$, respectively (*Chen et al., 2014*), where $\Delta\delta_X$, denotes the chemical shift differences between apo and SUMO1-bound SENP1 for nucleus X.

### Combination of replicate simulations

The three replicate simulations for each system were combined to ensure reproducibility of reported results. The principal component analysis was done after pooling the conformations (every other saved ones) from all the three simulations, using Cα coordinates of SENP1 residues 429–637. For

distributions of the cleft distance and loop movements, histograms were calculated using all the snapshots saved in the three replicate simulations. Chemical shifts were predicted separately for each simulation (using every 15th saved conformation in sim1 or sim2 or every 10th conformation in sim3), and the results from the three simulations were then averaged. For RMSF, analysis was done over non-overlapping 50-ns windows (of which there were 3, 3, and 6, respectively, in sim1, sim2, and sim3). The results were first averaged over these windows in each simulation, and then averaged again over the three replicate simulations. Finally, a community analysis was done separately for each replicate simulation.

## Acknowledgements

We thank Dr. Yuan Chen for introducing the SENP1 allostery problem to us. This work was supported by National Institutes of Health Grants GM058187 and GM118091.

## Additional information

### Funding

| Funder | Grant reference number | Author |
| --- | --- | --- |
| National Institutes of Health | GM058187 | Huan-Xiang Zhou |
| National Institutes of Health | GM118091 | Huan-Xiang Zhou |

The funders had no role in study design, data collection and interpretation, or the decision to submit the work for publication.

### Author contributions

JG, Conception and design, Acquisition of data, Analysis and interpretation of data, Drafting or revising the article; H-XZ, Conception and design, Analysis and interpretation of data, Drafting or revising the article

### Author ORCIDs

Huan-Xiang Zhou, http://orcid.org/0000-0001-9020-0302

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
