## [Decision Letter]

Thank you for submitting your article "Allosteric Activation of SENP1 by SUMO1 β-Grasp Domain Involves a Dock-and-Coalesce Mechanism" for consideration by *eLife*. Your article has been favorably evaluated by John Kuriyan (Senior Editor) and three reviewers, one of whom, Volker Dötsch, is a member of our Board of Reviewing Editors..

The reviewers have discussed the reviews with one another and the Reviewing Editor has drafted this decision to help you prepare a revised submission.

Summary:

The authors report molecular dynamics simulations to study the mechanism of allosteric activation of SENP1 by SUMO1 β-Grasp domain. The design of the simulations was motivated in part by previous NMR studies and in part by previous simulation study by the authors of a different system (PIN1). The mechanism that emerged from the simulations, referred to as a "dock-and-coalesce" mechanism, suggests that the binding of SUMO1 quenches fast (nanosecond) dynamics while stimulates slow (microsecond-millisecond) collective motions that couple distal regions together. In particular, the collective motion may facilitate structural rearrangements in the active site, therefore impacting the catalytic efficiency of SENP1.

Overall, the study is well motivated by a problem (allostery) of general significance. The proposed mechanism is qualitatively consistent with reported experimental data (distinct effects on fast and slow dynamics), and it provides novel physical insights into the allosteric activation of SENP1 and related systems (e.g., DUB). The mechanism also provides new clues to the design of small molecules that target SENP1 activation.

Essential revisions:

1) It is now becoming relatively easy to reach microsecond time scales in MD simulations, even for relatively large systems, and one should make an argument for why 200 ns were deemed appropriate to address the questions posed here since modern computational resources would allow simulations in excess of what is presented here.

2) It is also now common practice to reach statistical redundancy by running replicate simulations. To obtain statistical significance of the results a replication of the simulation would be important.

3) The figure caption of Figure 2 should mention that 2C maps the sampling onto the principal components. Otherwise, the figure is confusing.

4) It would help to indicate in Figure 2 where exactly the cleft distance was measured.

5) The comparison with the experimental chemical shift perturbations validates the simulations but doesn't quite help with the mechanistic picture. It would be useful to discuss a mechanistic model that explains the observed effects. In particular, how does triggering slow motions lead to reorganization of the active site that favors catalysis. Additional discussion that further clarifies this important aspect of activation would be beneficial to the clarity of the proposed mechanism.

---

## [Author Response]

*1) It is now becoming relatively easy to reach microsecond time scales in MD simulations, even for relatively large systems, and one should make an argument for why 200 ns were deemed appropriate to address the questions posed here since modern computational resources would allow simulations in excess of what is presented here.*

We now report results from three replicate simulations for each of the three systems. The lengths of these simulations were 0.3, 0.4, and 1.0 microseconds, accumulating a total of 5.1 microseconds for the three systems.

*2) It is also now common practice to reach statistical redundancy by running replicate simulations. To obtain statistical significance of the results a replication of the simulation would be important.*

As stated in the preceding reply, we have now run three replicate simulations for each system. The conformational and dynamic results reported in the revision are those reproduced in all the three independent simulations, and hence are statistically significant.

*3) The figure caption of Figure 2 should mention that 2C maps the sampling onto the principal components. Otherwise, the figure is confusing.*

We now explain how the distributions in Figure 2 were calculated to avoid confusion (subsection “Combination of Replicate Simulations”).

*4) It would help to indicate in Figure 2 where exactly the cleft distance was measured.*

Done.

*5) The comparison with the experimental chemical shift perturbations validates the simulations but doesn't quite help with the mechanistic picture. It would be useful to discuss a mechanistic model that explains the observed effects. In particular, how does triggering slow motions lead to reorganization of the active site that favors catalysis. Additional discussion that further clarifies this important aspect of activation would be beneficial to the clarity of the proposed mechanism.*

We agree that chemical shift perturbations, from either experiments or simulations, are only indicators of allosteric effects and by themselves do not provide mechanistic insight. That’s why the analyses of conformational and dynamic properties in the simulations are useful – they potentially lead to mechanistic explanations. As for the link between slow motions and catalytic activation, we have now made the discussion of our proposed mechanism somewhat tighter and more to the point (subsection “A dock-and-coalescence mechanism for SENP-catalyzed SUMO cleavage”, third paragraph and Figure 7 caption), but we feel a more detailed discussion of this link would be too speculative at this time.